# Exploratory Evaluation of Pre-Treatment Inflammation Profiles in Patients with Colorectal Cancer

**DOI:** 10.3390/diseases12030061

**Published:** 2024-03-20

**Authors:** Catalin Vladut Ionut Feier, Calin Muntean, Sorana D. Bolboacă, Sorin Olariu

**Affiliations:** 1First Discipline of Surgery, Department X-Surgery, “Victor Babeș” University of Medicine and Pharmacy, 2 Eftimie Murgu Sq., 300041 Timisoara, Romania; catalin.feier@umft.ro (C.V.I.F.); olariu.sorin@umft.ro (S.O.); 2First Surgery Clinic, “Pius Brinzeu” Clinical Emergency Hospital, 300723 Timisoara, Romania; 3Medical Informatics and Biostatistics, Department III-Functional Sciences, “Victor Babeș” University of Medicine and Pharmacy, 2 Eftimie Murgu Sq., 300041 Timisoara, Romania; 4Department of Medical Informatics and Biostatistics, “Iuliu Hațieganu” University of Medicine and Pharmacy Cluj-Napoca, 6 Louis Pasteur Street, 400349 Cluj-Napoca, Romania; sbolboaca@umfcluj.ro

**Keywords:** colorectal cancer, inflammation biomarkers, systemic immune–inflammation index, prognosis

## Abstract

In light of the elevated incidence and consequential prognostic implications associated with colorectal cancer, a comprehensive investigation into the impact exerted by inflammatory status on patient management becomes imperative. A retrospective study spanning 7 years was conducted, involving the retrospective collection of data on colorectal cancer patients undergoing surgical intervention. We evaluated six inflammation ratios derived from complete peripheral blood counts. A thorough analysis of these markers’ prognostic capacity was conducted, revealing that patients who died postoperatively displayed significantly higher preoperative Aggregate Index of Systemic Inflammation—AISI (*p* = 0.014) and Systemic Inflammation Response Index—SII (*p* = 0.0197) levels compared to those with successful discharge. Noteworthy variations in neutrophil-to-lymphocyte ratio (*p* = 0.0103), platelet-to-lymphocyte ratio (*p* = 0.0041), AISI (*p* < 0.001), and SII (*p* = 0.0045) were observed in patients necessitating postoperative Intensive Care Unit (ICU) monitoring. Furthermore, patients with complications, such as an intestinal fistula, exhibited significantly elevated AISI (*p* = 0.0489). Inflammatory biomarkers stand out as valuable prognostic tools for colorectal cancer patients, offering potential assistance in predicting their prognosis.

## 1. Introduction

Colorectal cancer (CRC) poses a significant health challenge, ranking as the third most common cancer globally according to Globocan 2020, with more than 1.9 million new cases of colorectal cancer and more than 930,000 deaths due to colorectal cancer estimated to have occurred worldwide in 2020 [1]. Over two decades (from 1999 to 2019), Romania has witnessed a concerning increase in CRC incidence and mortality rates, reaching 17.74/100,000 inhabitants [2]. Despite considerable advancements in CRC treatment, long-term survival remains suboptimal, with approximately 60% of patients experiencing a 5-year survival rate post-curative resection [3].

Surgical intervention remains essential in CRC treatment, with modern technologies markedly improving the outcomes and overall survival. However, precise preoperative tumor staging remains a challenge, emphasizing the crucial role of identifying prognostic biomarkers in optimizing patient management [4].

The investigation of biological markers has become pivotal in understanding systemic inflammation and its role in cancer progression. Various ratios, including neutrophil-to-lymphocyte (NLR), platelet-to-lymphocyte (PLR), and monocyte-to-lymphocyte (MLR), are extensively studied in colorectal cancer (CRC) to offer insights into patients’ inflammatory status and predict disease progression [2,3]. Notably, studies present that an elevated NLR (> 5) has been linked to a decrease in overall survival [5], while increased PLR values (>140) have been correlated with higher postoperative morbidity rate [6]. Hemoglobin, NLR, and platelet count serve as additional biomarkers, offering insights into the pre-surgery tumor status. Furthermore, emphasizing the significance of inflammatory indices such as AISI (aggregate index of systemic inflammation), SIRI (systemic inflammation response index), and SII (Systemic Immune–Inflammation Index) have the potential to predict outcomes in CRC [7,8].

Understanding the intricate interplay between systemic inflammation and CRC progression is essential for refining prognostic assessments and therapeutic strategies. The identified biomarkers offer valuable information, underscoring the necessity for personalized approaches based on the inflammatory status of CRC patients.

Our study aimed to evaluate the preoperative inflammation status of patients with colorectal cancer, identify associations between this status and several parameters, and evaluate, if any, have a possible impact on postoperative prognosis.

## 2. Materials and Methods

We conducted our study according to the Declaration of Helsinki and received approval from the “Pius Brinzeu” Clinical Emergency Hospital Ethical Commission prior to collecting data (approval no. 434/29 January 2024).

### 2.1. Design and Settings

To reach our aim, we conducted an observational, analytical study with a retrospective collection of data from medical charts. Patients undergoing surgical intervention for primary colorectal cancer treatment at the First Surgery Clinic of the “Pius Brinzeu” Clinical Emergency Hospital, Romania, represented the eligible population. The time frame of our study covers 7 years, from 1 January 2016 to 31 December 2022.

Figure 1 presents the flow of the applied methods. The histopathological examination was the gold-standard diagnosis in our study. Medical charts of patients were the source of raw data.

We excluded patients with chemotherapy or radiotherapy before the surgery because their effects on systemic inflammation (either increased or decreased) were observed for months after treatment, and Zou et al. mentioned that adjuvant chemotherapy appeared to be more effective in patients with CRC with high NLR or PLR [9,10,11]. Similarly, SARS-CoV-2 infection is associated with a systemic inflammatory response, besides the inflammation associated with the presence of colorectal cancer [12]; in other words, patients with a history of this infection prior to the intervention or patients who developed an infection during the hospital stay were excluded.

We included patients with primary tumors localized from the cecum to the rectosigmoid junction. Four categories were used for tumor location: right colon (tumors at the cecum, ascending colon, hepatic flexure), transverse colon, and left colon (tumors at the splenic flexure, descending colon, sigmoid colon, and rectosigmoid junction).

### 2.2. Data Analysis

We used centrality and dispersion indicators to summarize the characteristics of the sample. We reported the mean and standard deviation for normally distributed data (Shapiro–Wilk test). For non-normally distributed quantitative variables, we used the median and interquartile range [Q1 to Q3]. We report qualitative variables as numbers and percentages. We applied the Mann–Whitney test to compare two independent groups and the Kruskal–Wallis test to compare two groups for quantitative data that violated, in at least one sub-group, the normal distribution. We applied the Chi-squared test or Fisher’s exact test to compare groups in the case of attribute data.

We used Statistica (v.13.5, TIBCO Software Inc., Palo Alto, CA, USA) to conduct our exploratory statistical analysis. We obtained the graphical representations with Jasp (v. 0.18.3.0, available at: https://jasp-stats.org/ (accessed on 10 January 2024)). We used a significance level of 5% and considered the *p*-values smaller than 0.05 as statistically significant.

## 3. Results

We evaluated 282 patients aged from 26 to 97 years with a balance among sex. As expected, due to restricted access to clinical care, the number of cases was lower during the COVID-19 pandemic (39 in 2016, 43 in 2017, 65 in 2018, 41 in 2019, 31 in 2020, 29 in 2021, and 34 in 2022).

### 3.1. Deceased vs. Alive

The patients who died during hospitalization were older, had a CHARLSON score higher than 3, most frequently had the surgery in emergency, had postoperative complications, and needed intensive care in the ICU (Table 1).

Patients who died showed significantly less lymphocytes and neutrophils, and higher values of AISI and SII than those who survived (Table 2, Figure 2).

### 3.2. Elected vs. Emergency Surgery

Patients who had an unplanned surgery were similar to those who had a planned surgery in terms of age (*p*-value = 0.1008), sex (*p*-value = 0.2468), living in rural areas (*p*-value = 0.2310), hospitalization stay (*p*-value = 0.0540), presence of relapse (*p*-value = 0.4353), post-surgery complications (*p*-value = 0.2278), and CHARLSON score (*p* = 0.3185). A statistically significant smaller percentage of patients with unplanned surgery had a curative intervention (104 (77%) vs. 128 (87.1%), *p*-value = 0.0275), significantly higher frequency of lymphovascular invasion (70 (56.9%) vs. 58 (42%); *p*-value= 0.0164), with a lower percentage of patients in stage I (4 (3.3%) vs. 19 (13.7%)) and a higher percentage of patients in stage IV (24 (19.5%) vs. 18 (12.9%)) (χ^2^ = 11.2, *p*-value = 0.0105). Patients with emergency surgery showed a different inflammation profile compared to those with planned surgery (Table 3, Figure 3).

### 3.3. Stage of the Disease

The AISI varied from 62 to 23,044, with statistically significant differences in patients with different stages of the disease (Kruskal–Wallis test; *p*-value = 0.0313). The lowest values were observed in patients with stage I (323 [266.5 to 710.5], 20) and the highest values in patients with stage IV (1082 [433.5 to 1975], 23), with a statistically significant difference in posthoc analysis (*p*-value = 0.0328) (Figure 4).

Patients with a regional stage of disease (serosa invasion, pT = 4) are similar to those without serosa penetration in terms of demographic characteristics. The number of neutrophils (6540 [5290 to 8570], 53 vs. 5360 [4000 to 7340], 99; Mann–Whitney test: 2.69 (0.0073)) and platelets (6540 [5290 to 8570], 53 vs. 5360 [4000 to 7340], 99; Mann–Whitney test: 2.98 (0.0029)) showed statistically significant different values compared to those without a regional stage of disease. Four of the evaluated inflammation ratios demonstrated higher values for patients with serosa penetration than those without (Table 4).

### 3.4. Relapse, Early Complications and Post-Surgery Intensive Care Admission

Twenty-seven patients (9.6%) in our cohort were with relapse. Patients with relapse were younger (60 [56.5 to 67.5], 27 vs. 69 [60.5 to 75], 255; *p*-value = 0.0034) and had more hospitalization stays (16 [11.5 to 23], 27 vs. 13 [10 to 17], 255; *p*-value = 0.0058). The evaluated ratios were similar between those with and without relapse (*p*-values > 0.15).

Seventeen (6.0%) patients had a fistula and exhibited three times higher value of AISI than those without a fistula (2373 [971 to 3675], 9 vs. 743 [332 to 1344.5], 156; *p*-value = 0.0489).

Post-surgery, thirty-five patients were admitted to the ICU (12.4%). Patients with post-surgery ICU were older (72 years [64 to 77.5], *n* = 35 vs. 67 years [59 to 74], *n* = 247; *p*-value = 0.0247), have statistically significant NLR (6.3 [3.4 to 8.3], *n* = 13 vs. 3.5 [2.3 to 5.2], *n* = 152; *p*-value = 0.0103), PLR (250 [174.5 to 330.5], *n* = 35 vs. 177 [134 to 261.5], *n* = 247; *p*-value = 0.0041), AISI (2817 [1169 to 4575], *n* = 13 vs. 646.5 [323.5 to 1313.3], *n* = 152; *p*-value = 0.0001), and SII (2207 [1065 to 3846], *n* = 13 vs. 1061.5 [611.5 to 1711.3], *n* = 152; *p*-value = 0.0045).

## 4. Discussion

The evaluated inflammation ratios showed different patterns on specific outcomes. Specifically, AISI and SII showed higher values in deceased patients, NLR, PLR, AISI, and SII exhibited elevated values, and MLR and SIRI showed lower values in patients who underwent emergency surgery. Furthermore, AISI showed elevated values in patients with a higher stage of disease, and NLR, PLR, AISI, and SII proved statistically significantly higher in patients with a regional stage compared to those without a regional stage.

The timeframe of our study includes the COVID-19 pandemic, during which surgical interventions decreased by over 50% in the first year compared to the pre-pandemic period, in line with global trends [13,14,15]. To mitigate potential interference with results due to the impact of SARS-CoV-2 infection on the inflammatory system [16,17,18], patients with a history of SARS-CoV-2 infection or those who acquired the infection during hospitalization were excluded from the study.

Our cohort had an average age of 68 years, predominantly male, consistent with the literature findings on colorectal cancer incidence. Siegel et al. reported an incidence of 40.7 per 100,000 individuals for colorectal cancer, with a higher frequency in males [19]. The decline in colorectal cancer mortality since 1980 can be attributed to improved screening methods, such as endoscopic detection of colonic polyps and minimally invasive surgical interventions [20]. Incidence peaks in the fourth to sixth decades of life, with age-specific rates increasing with each subsequent decade [21]. In our study, deceased patients were statistically significantly older, underwent emergency surgeries at a higher proportion, and had a higher CHARLSON index over 3 (Table 1). Associated pathologies, acid-base, electrolyte imbalances, and complications requiring emergency surgical intervention contributed to a poorer prognosis [22]. Deceased patients also required intensive care monitoring at a significantly higher proportion (*p* < 0.0001) and experienced an intestinal fistula as a postoperative complication (*p* < 0.0001, Table 1). The incidence of stage IV disease was higher in deceased individuals compared to survivors (25.7% vs. 13.4%, Table 1). Our findings align with the scientific literature emphasizing the impact of negative prognostic factors for colorectal cancer patients undergoing surgical intervention [23,24]. The inflammatory status has been long recognized as a pivotal factor in the progression of cancer in patients [25]. The emergence of cancer is often linked with persistent inflammatory conditions, with infections contributing to over 15% of malignancies [26]. The findings of our study reveal substantial disparities among patients with specific outcomes. Notably, the count of lymphocytes exhibits a significant contrast, showing a decline in non-survivors compared to survivors (Table 2). Hence, the hematologic equilibrium and response of patients exert a significant influence on postoperative outcomes and overall prognosis [27]. Similarly, the count of neutrophils displays a marked elevation among deceased individuals relative to survivors (Table 2), while the numerical increase in NLR among deceased patients compared to survivors showed only a tendency to statistical significance (0.01 < *p*-value < 0.10).

Several studies have underscored the utility of NLR as a prognostic indicator in CRC patients prior to surgery [28,29], although a universally accepted threshold remains undetermined. Despite the fact that MLR has been associated with less favorable prognoses [30,31], our study shows similar values in patients who died than in those who survived (Table 2). The mean NLR value for deceased patients stood at 4.47 in our study, a result in line with scientific data with a threshold of 3.3 [32]. Additionally, Ding et al. showed that an elevated preoperative NLR (>4) emerges as an independent predictor of poorer survival in CRC patients [33]. Other studies have yielded similar outcomes, emphasizing the significance of this metric. For instance, Shibutani et al. demonstrated that a preoperative NLR > 2.5 significantly forecasts poorer cancer-specific survival in CRC patients [34]. Within this study, a notable increase in AISI is also evident among patients with a fistula, underscoring once again the link between imbalances in the inflammatory response and adverse prognostic factors [23,24].

As expected, the examination of the relationship between biomarkers and the regional stage of colorectal cancer unveils notable disparities. NLR demonstrates a significant surge in patients with regional stages compared to those with non-regional stages (Table 4). Similarly, PLR shows a substantial increase in patients with regional stages, and AISI and SII exhibit analogous trends, with significant elevations in patients with regional stages (Table 4). AISI showed a high variability (from 62 to 23,044), with marked distinctions between patients with varying disease stages (Kruskal–Wallis test; *p*-value = 0.0313). The lowest values were observed in patients with stage I, while the highest values were recorded in patients with stage IV (*p*-value = 0.0328), the reaction of the organism increasing with the stage of the disease.

Patients with the regional stage of the disease (lymphovascular invasion, pT = 4) are similar to those without serosa penetration in terms of demographic characteristics but showed significantly higher numbers of neutrophils (*p* = 0.0073) and platelets (*p* = 0.0029). Four of the evaluated inflammation markers showed higher values in patients with serosa penetration. Similarly, an extensive study highlighting significant associations between NLR, derived neutrophil-to-lymphocyte ratio (dNLR), and T stage in colorectal cancer has been reported in the scientific literature, suggesting their potential clinical relevance [22]. Additionally, MLR shows an inverse relationship with the T stage. Hu et al. [35] reported that SII is also associated with poor histological differentiation, larger tumor sizes, and advanced T, N, and M stages, validating the hypothesis that increased inflammatory response could promote tumor proliferation, progression, and metastasis [36]. The combined evaluation of NLR and SII reflects the clinical utility of these individual ratios [37]. While NLR proves valuable in identifying patients with positive lymph nodes [38], MLR does not reach the reliability demonstrated by NLR or dNLR [39]. Including PLR in TNM staging could enhance prognostic capacity [40].

It is pertinent to highlight that a derived neutrophil-to-lymphocyte ratio (dNLR) equal to or greater than 3.125 effectively doubles the risk of mortality, while each year of advancing age escalates the risk by 4%. Notably, both neutrophil-to-lymphocyte ratio (NLR) and monocyte-to-lymphocyte ratio (MLR) maintain significant associations with the localized stage of the tumor, irrespective of gender and age. An initial NLR exceeding 3.105 prior to treatment amplifies the likelihood of an advanced T stage, whereas each increment in the MLR ratio diminishes this probability. Moreover, an NLR surpassing 4.255 significantly triples the likelihood of metastasis [22]. Several studies corroborate the potential of inflammatory markers in prognostic prediction and risk assessment in colorectal cancer [41,42,43,44,45]. Results regarding the type of surgical intervention—elective or emergency—reveal significant insights. The number of lymphocytes, although not statistically significant (Table 3), shows a marginal decrease in patients with emergency interventions compared to those with scheduled interventions. On the other hand, the number of monocytes is significantly lower in patients with emergency interventions compared to those with elective interventions (Table 3), showing the different baseline biomarkers between these two groups. This study has highlighted significant differences in inflammatory markers between patients undergoing emergency and elective surgeries for colorectal cancer. NLR shows a significant increase in emergency surgeries, while the monocyte-to-lymphocyte ratio indicates a substantial rise in emergency surgery patients (Table 3). Patients undergoing emergency surgeries exhibit a significantly increased PLR, AISI, and SII (Table 3). These findings highlight the baseline response of the body to the presence of CRC that could explain the burn and complexity of the disease at presentation. Additionally, since patients admitted to postoperative intensive care had a significantly higher mortality rate, it is noteworthy that they presented significantly higher values for NLR (*p* = 0.0103), PLR (*p* = 0.0041), AISI (*p* = 0.0001), and SII (*p* = 0.0045).

Thus, following the presentation of our study results and their link with the relevant literature, it can be concluded that a significant variation in inflammatory status parameters can be employed in the prognostic analysis of these patients. Stotz et al. [45] underscore the preoperative predictive value of MLR and its correlation with shortened long-term survival. Huang et al. [46] present, in a meta-analysis, the predictive superiority of PLR, demonstrating its association with reduced survival and increased recurrence. However, SII takes center stage as the most effective predictor of long-term survival outcomes, surpassing NLR and PLR. Its comprehensive reflection of inflammatory and immune responses positions SII in a key role in prognostic assessments for colorectal cancer, offering valuable perspectives on potential implications [7,8,47,48]. These findings collectively emphasize the intricate connection between inflammatory markers and outcomes in the context of colorectal cancer patients, providing crucial insights into prognostic implications. Furthermore, the correlation between age, postoperative intensive care unit admission, and elevated levels of NLR, PLR, AISI, and SII underscores the complex impact of these markers, adding depth to the understanding of their clinical relevance in the postoperative setting [8,22,23,24,48,49].

Our findings highlight significant patterns in inflammatory markers and the regional stage of colorectal cancer, providing a more detailed understanding of potential clinical implications. Through a careful analysis of the data obtained in this study, promising perspectives for clinical management and prognosis of patients with colorectal cancer are outlined.

### Study Limitations

The conclusions drawn from our study demand meticulous consideration, as they are accompanied by several limitations that require careful acknowledgment. First, the retrospective nature of our data collection precluded our ability to adequately control for potential confounding variables. This limitation may have resulted in the exaggeration or underestimation of the associations identified, particularly considering the omission of influential factors such as smoking, inflammatory conditions, medication usage, genetic predispositions, diabetes, and obesity. Second, it is essential to acknowledge the potential for misclassification of outcomes, notably the staging of tumors (T stage and M stage). Variations in imaging techniques and interpretations by different healthcare professionals may have contributed to a misclassification, potentially impacting the perceived clinical utility of the investigated ratios. To address these limitations and enhance the robustness of future research, a prospective approach with standardized protocols for outcome assessment is warranted. Furthermore, comprehensive consideration of potential confounding variables, including those omitted in our study, is imperative. Additionally, investigating the dynamic changes in ratios before and after interventions could yield valuable insights into their clinical relevance. However, the significant contributions of this study to understanding the role of inflammation in colorectal cancer underscore the importance of continued research to identify and validate relevant inflammatory markers with both prognostic and therapeutic potential.

## 5. Conclusions

Our results revealed distinct trends of evaluated inflammation ratios linked to various outcomes. Notably, AISI and SII demonstrated elevated levels among deceased patients, while NLR, PLR, AISI, and SII exhibited heightened values among patients undergoing emergency surgery. Moreover, AISI levels were found to be elevated in patients with advanced disease stages, and statistically significant increases in NLR, PLR, AISI, and SII were observed in patients with regional disease stages compared to those without.

Moving forward, adopting a prospective approach with standardized protocols and accounting for comprehensive confounding variables are imperative to strengthen future research endeavors. Despite the limitations, our study underscores the importance of further exploration to validate inflammatory markers’ prognostic and therapeutic potential in colorectal cancer management.

## Figures and Tables

**Figure 1 diseases-12-00061-f001:**
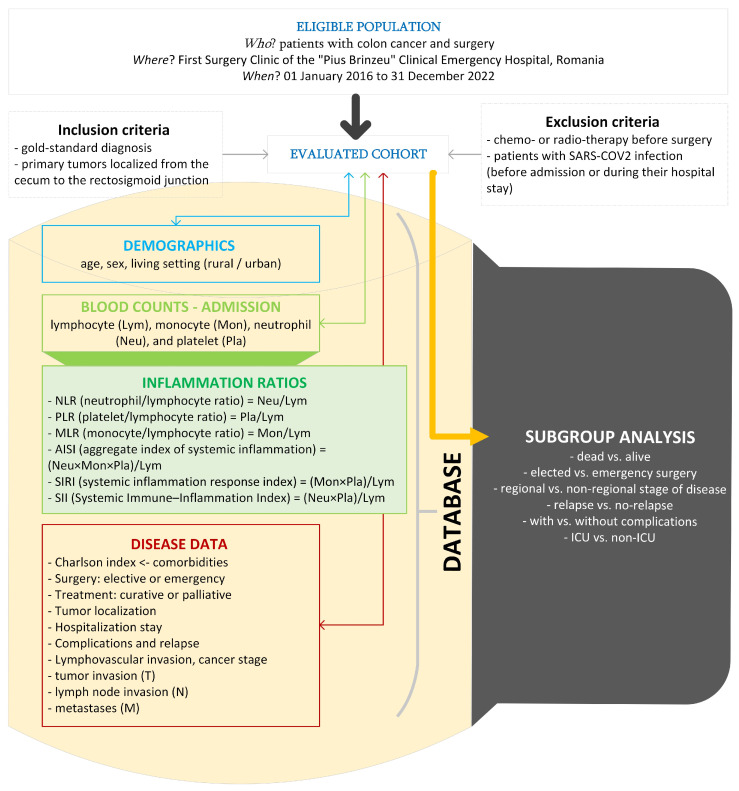
The setting, raw, and derived data in our study (drawn with Microsoft Visio, v. 16.0 2019, Microsoft Corporation, Redmond, WA, USA).

**Figure 2 diseases-12-00061-f002:**
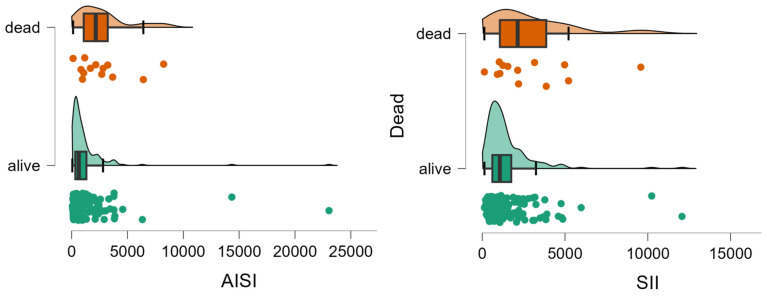
Distribution of AISI and SII values by group. The circles represent the raw data, the box shows the median (middle line), first and third quartiles (box), respectively, and minimum and maximum values (whiskers).

**Figure 3 diseases-12-00061-f003:**
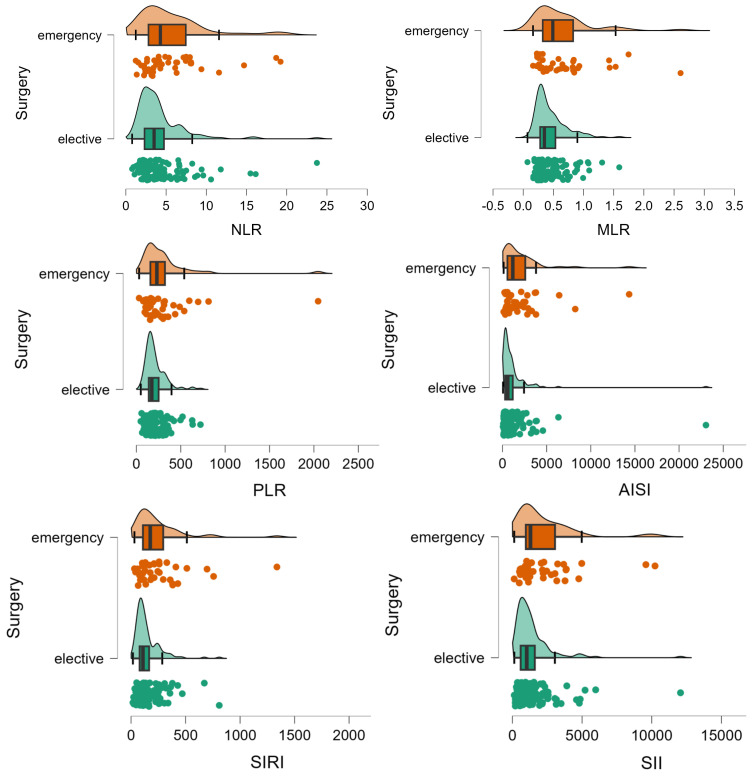
Distribution of inflammation ratios on patients with (elective) and without elective surgery (emergency). The circles represent the raw data, the box showed the median (middle line), first and third quartiles (box), respectively minimum and maximum values (whiskers).

**Figure 4 diseases-12-00061-f004:**
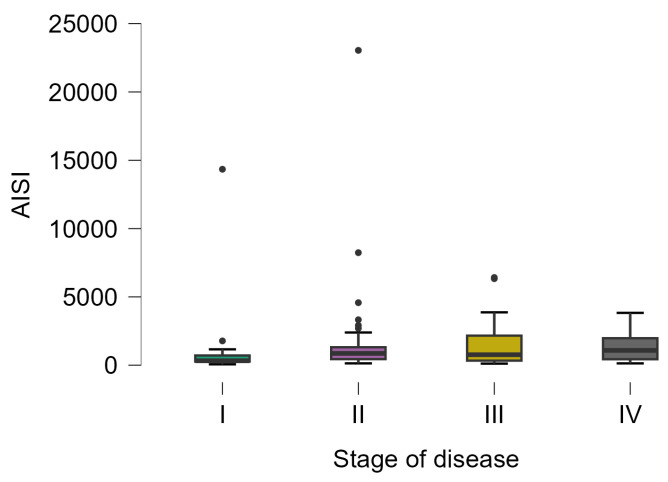
Distribution of AISI on patients with different disease stages. The circles represent the raw data, and the box shows the median (middle line), first and third quartiles (box), respectively, and minimum and maximum values (whiskers).

**Table 1 diseases-12-00061-t001:** Cohort features and comparison between deceased and alive patients.

Characteristic	All, *n* = 282	Dead, *n* = 35	Alive, *n* = 247	Stat. (*p*-Value)
Age, years ^a^	68 [60 to 74], 282	73 [66 to 78], 35	67 [59 to 73], 247	2.86 (0.0042)
Sex, men ^b^	148 (52.5)	23 (65.7)	125 (50.6)	2.81 (0.0939)
Rural ^b^	99 (35.2)	11 (31.4)	88 (35.8)	0.25 (0.6147)
CHARLSON > 3 ^b^	111 (39.4)	24 (68.6)	87 (35.2)	14.28 (0.0002)
Emergency ^b^	135 (47.9)	30 (85.7)	105 (42.5)	22.93 (<0.0001)
Curative surgery ^b^	232 (82.3)	24 (68.6)	208 (84.2)	5.14 (0.0234)
Localization ^b^				n.a. (0.15406)
Right	83 (29.4)	13 (37.1)	70 (28.3)	
Left	162 (57.4)	17 (48.6)	145 (58.7)	
Transverse	26 (9.2)	1 (2.9)	25 (10.1)	
Relapse ^b^	27 (9.6)	2 (5.7)	25 (10.1)	n.a. (0.3188)
Complications ^b^	16 (5.7)	9 (25.7)	7 (2.8)	29.99 (<0.0001)
Postop ICU ^b^	35 (12.4)	22 (62.9)	13 (5.3)	93.54 (<0.0001)
Stage				n.a. (0.0571)
I	23 (8.2)	0 (0)	23 (9.3)	
II	87 (30.9)	8 (22.9)	79 (32)	
III	110 (39)	13 (37.1)	97 (39.3)	
IV	42 (14.9)	9 (25.7)	33 (13.4)	
Lymphatic invasion ^b^	128 (49)	18 (60)	110 (47.6)	1.63 (0.2019)
pT ^b^				n.a. (0.2893)
1	8 (3.1)	0 (0)	8 (3.5)	
2	18 (7)	1 (3.4)	17 (7.5)	
3	123 (48)	11 (37.9)	112 (49.3)	
4	107 (41.8)	17 (58.6)	90 (39.6)	
pN ^b^				1.62 (0.4443)
0	117 (45.9)	11 (37.9)	106 (46.9)	
1	74 (29)	8 (27.6)	66 (29.2)	
2	64 (25.1)	10 (34.5)	54 (23.9)	
pM ^b^	40 (15.4)	8 (26.7)	32 (13.9)	3.32 (0.0686)
Hospitalization, days ^a^	13 [10 to 17], 282	9 [3 to 19.5], 35	13 [10 to 17], 247	−2.74 (0.0062)
Post-surgery, days ^a^	10 [8 to 14], 282	6 [2 to 18], 35	10 [8 to 13], 247	−2.58 (0.0099)

Data are reported as median [Q1 to Q3], where Q is the quartile (^a^) or no. (%) (^b^) according to the type of raw data. Stat. = statistics of the test; *p*-value = probability associated with the statistics of the test. Mann–Whitney test was used to compare two independent groups for quantitative data (^a^). Chi-squared or Fisher’s exact test was used to compare qualitative data (n.a. = not applicable to the Fisher’s exact test) (^b^). ICU = intensive care unit; T = tumor invasion; N = lymph node invasion; M = presence of metastases.

**Table 2 diseases-12-00061-t002:** Evaluated biomarkers and associated ratios by groups.

Marker	All, *n* = 282	Dead, *n* = 35	Alive, *n* = 247	Stat. (*p*-Value)
Lymphocytes	1600 [1212.5 to 2100], 282	1420 [795 to 1930], 35	1600 [1290 to 2100], 247	−1.99 (0.0461)
Monocytes	425 [200 to 720], 282	300 [200 to 905], 35	450 [300 to 700], 247	−1.32 (0.1883)
Platelets	303,800 [242,000 to 384,000], 282	306,000 [202,000 to 407,000], 35	303,600 [245,000 to 383,000], 247	−0.34 (0.7331)
Neutrophils	5760 [4380 to 7830], 165	7180 [5720 to 10,500], 13	5605 [4100 to 7697.5], 152	2.32 (0.0202)
NLR	3.63 [2.41 to 5.44], 165	4.47 [3.39 to 11.58], 13	3.57 [2.355 to 5.3225], 152	1.73 (0.0837)
MLR	0.28 [0.15 to 0.47], 282	0.25 [0.129 to 0.5515], 35	0.281 [0.159 to 0.4495], 247	−0.39 (0.6934)
PLR	185.5 [137 to 279.25], 282	250 [134.5 to 332], 35	183 [137 to 257.5], 247	1.56 (0.1182)
AISI	765 [332 to 1546], 165	2162 [1082 to 3233], 13	646.5 [329.75 to 1325.75], 152	3.19 (0.0014)
SIRI	83 [44 to 139], 281	72 [32.75 to 205.75], 34	83 [46 to 132.5], 247	−0.18 (0.8606)
SII	1070 [645 to 2009], 165	2140 [1065 to 3868], 13	1061.5 [612.75 to 1754.5], 152	2.33 (0.0197)

Data are reported as median [Q1 to Q3], where Q is the quartile. Mann–Whitney test was used to compare the two independent groups.

**Table 3 diseases-12-00061-t003:** Inflammation profile of patients with or without an elected surgery.

Marker	Emergency Surgery, *n* = 135	Elected Surgery, *n* = 147	Stat. (*p*-Value)
Lymphocytes	1570 [1055 to 2120], 135	1650 [1310 to 2055], 147	−1.29 (0.1986)
Monocytes	300 [200 to 595], 135	550 [400 to 790], 147	−6.00 (<0.0001)
Platelets	310,000 [239,500 to 381,500], 135	300,000 [247,000 to 384,000], 147	0.22 (0.8225)
Neutrophils	6840 [5000 to 8680], 41	5525 [4087.5 to 7440], 124	2.25 (0.0243)
NLR	4.29 [2.8 to 7.5], 41	3.51 [2.3 to 4.7], 124	2.26 (0.0237)
MLR	0.182 [0.1 to 0.3], 135	0.319 [0.2 to 0.5], 147	−5.16 (<0.0004)
PLR	197 [135 to 298], 135	175 [138 to 248], 147	3.15 (0.0016)
AISI	1165 [559 to 2580], 41	599.5 [293.5 to 1162.3], 124	−4.05 (0.0001)
SIRI	59.5 [34 to 118], 134	96 [61 to 155], 147	−4.05 (0.0001)
SII	1297 [966 to 3058], 41	1030 [594.8 to 1609.5], 124	2.72 (0.0064)

Data are reported as median [Q1 to Q3], where Q is the quartile. Mann–Whitney test was used to compare two independent groups. The number after the bracket represents the eligible number of patients.

**Table 4 diseases-12-00061-t004:** Inflammation ratios by regional stage of disease.

Ratio	Regional Stage of Disease	No-Regional Stage of Disease	Stat. (*p*-Value)
NLR	3.83 [3.13 to 6.31], 53	3.19 [2.18 to 4.52], 99	2.98 (0.0029)
PLR	215 [144.5 to 305], 107	170.5 [128 to 232], 150	2.64 (0.0084)
AISI	1082 [506 to 2373], 53	570 [287 to 1082], 99	3.68 (0.0002)
SII	1449.5 [990.25 to 2360.25], 52	966 [543.5 to 1492.5], 99	3.55 (0.0004)

## Data Availability

The datasets used and/or analyzed during the current study are available from the corresponding author upon reasonable request.

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
