# Peer review of "Exploratory Evaluation of Pre-Treatment Inflammation Profiles in Patients with Colorectal Cancer"

_diseases, 2024, doi:10.3390/diseases12030061_

Round 1

Reviewer 1 Report

Comments and Suggestions for Authors

Abstract

·         No comments.

Introduction

·         Emphasize the escalating incidence and mortality of colorectal cancer (CRC) in Romania more vividly by comparing it with global statistics to underline the urgency and relevance of the study.

·         Provide a more detailed rationale for choosing specific biomarkers like NLR, PLR, MLR, and others. Mention how these markers have been underexplored or inconsistently reported in CRC and propose how your study aims to bridge this knowledge gap.

·         Since the study includes the pandemic period, you could include a small discussion about the pandemic effects in oncology that were found in Romania. Your study could benefit from the following references:

  Porosnicu, T.M.; Sirbu, I.O.; Oancea, C.; Sandesc, D.; Bratosin, F.; Rosca, O.; Jipa, D.; Boeriu, E.; Bandi, S.S.S.; Pricop, M. The Impact of Therapeutic Plasma Exchange on Inflammatory Markers and Acute Phase Reactants in Patients with Severe SARS-CoV-2 Infection. Medicina 2023, 59, 867. https://doi.org/10.3390/medicina59050867

Methods

·         Patient selection criteria: Provide more comprehensive details on inclusion and exclusion criteria, including rationale for excluding patients with prior chemotherapy or radiotherapy and how these criteria ensure the study population accurately reflects the target demographic.

·         Make a separate subsection in the Materials and Methods section with inclusion and exclusion criteria and the rationale for those.

Results

·         Improve data presentation by adding graphical representation of your choice.

Discussion

·         Limitations and future directions are not found in the manuscript. Explicitly discuss the limitations of your study, including the retrospective design and any potential biases. Suggest specific future research directions, such as prospective studies, to validate your findings and explore the mechanisms behind the observed associations.

Conclusions

·         Relevant and sufficient.

Comments on the Quality of English Language

English language is appropriate. Major corrections are not needed.

Author Response

Q1. Introduction

  1. Emphasize the escalating incidence and mortality of colorectal cancer (CRC) in Romania more vividly by comparing it with global statistics to underline the urgency and relevance of the study.
  2. Provide a more detailed rationale for choosing specific biomarkers like NLR, PLR, MLR, and others. Mention how these markers have been underexplored or inconsistently reported in CRC and propose how your study aims to bridge this knowledge gap.
  3. Since the study includes the pandemic period, you could include a small discussion about the pandemic effects in oncology that were found in Romania.
  4. Your study could benefit from the following references [Porosnicu, T.M.; Sirbu, I.O.; Oancea, C.; Sandesc, D.; Bratosin, F.; Rosca, O.; Jipa, D.; Boeriu, E.; Bandi, S.S.S.; Pricop, M. The Impact of Therapeutic Plasma Exchange on Inflammatory Markers and Acute Phase Reactants in Patients with Severe SARS-CoV-2 Infection. Medicina 2023, 59, 867. https://doi.org/10.3390/medicina59050867].

Response: Thank you very muh for your comments and suggestion, we have made the modification accordingly to your requests

Q2. Methods

  1. Patient selection criteria: Provide more comprehensive details on inclusion and exclusion criteria, including rationale for excluding patients with prior chemotherapy or radiotherapy and how these criteria ensure the study population accurately reflects the target demographic.
  2. Make a separate subsection in the Materials and Methods section with inclusion and exclusion criteria and the rationale for those.

Response: The methods is now summarized into a chart. We clearly identified inclusion and exclusion criteria. Furthermore, we explained in the text why we chose the specific exclusion citeria

Q3. Results

Improve data presentation by adding graphical representation of your choice.

Response: Done. New graphical representation were included in the revised manuscript.

Q4. Discussion

Limitations and future directions are not found in the manuscript. Explicitly discuss the limitations of your study, including the retrospective design and any potential biases. Suggest specific future research directions, such as prospective studies, to validate your findings and explore the mechanisms behind the observed associations.

Response: We have dramatically change the discussion part of the article, therefore we have opted for writing a paragraph which presents out study`s limitation

Q5. Conclusions

Relevant and sufficient.

Response: Thank you for your appreciation.

Reviewer 2 Report

Comments and Suggestions for Authors

1. The authors only collected the data from patients with colon cancer to perform the analysis. However, why did they describe colorectal cancer? Please check.

2. It’s suggested to add a flowchart to show how they designed the study.

3. The limitations of study should be discussed deeply.

4. Why did they choose the factors, such as neutrophil-to-lymphocyte (NLR), platelet-to-lymphocyte (PLR), and monocyte-to-lymphocyte (MLR) in this study? The authors mentioned that previous studies have extensively studied these factors in predicting disease progression. Thus, what are the new findings in this study?

5. It’s suggested to perform ROC analysis to test the effect of these factors in predicting disease progression.

6. How did the authors choose the best cut-off values of the factors, like NLR?

7. The conclusion is superficial. Indeed, it is evident that the inflammatory status, such as NLR, MLR, PNI, AISI, and SII, hold significant relevance in the management of patients undergoing surgical treatment for colorectal cancer. However, how did we use the standard?

8. It’s suggested to check the grammar and typo errors.

Comments on the Quality of English Language

It’s suggested to check the grammar and typo errors.

Author Response

Q1. The authors only collected the data from patients with colon cancer to perform the analysis. However, why did they describe colorectal cancer? Please check.

Response: Yes, we described the colorectal because the reported data also refer rectal cancer.Moreover we included patients who underwent  surgical intervention for a malign tumor located from cecum cu rectosigmoid jonction, which includes a part o the rect as well.

Q2. It’s suggested to add a flowchart to show how they designed the study.

Response: Thank you for your suggestion. We added a flow chart that summarizes the study design.

Q3. The limitations of study should be discussed deeply.

Response: We have dramatically change the discussion part of the article, therefore we have opted for writing a paragraph which presents out study`s limitations

Q4. Why did they choose the factors, such as neutrophil-to-lymphocyte (NLR), platelet-to-lymphocyte (PLR), and monocyte-to-lymphocyte (MLR) in this study? The authors mentioned that previous studies have extensively studied these factors in predicting disease progression. Thus, what are the new findings in this study?

Response: Since no consensus is reported in the scientific literature, we tested how it is a work on our sample. It is also valuable to secondary studies, such as meta-analysis.

Q5. It’s suggested to perform ROC analysis to test the effect of these factors in predicting disease progression.

Response: Thank you for your suggestion. ROC analysis is correctly applied in the presence of gold standard diagnostic so it is appropriate only for regional stage of disease and if we p-phishing enough we can find some models with statistical significance. However, the performances of the models are limited (AUCs around 0.6) so without clinical relevance as already reported in the scientific literature (doi: 10.2174/1386207323666201020111946 & doi: 10.3390/diagnostics11030566 & doi: 10.3390/math8101741). As statistical exercise is good, but such analysis was outside the aim of our study and the approved protocol.

Q6. How did the authors choose the best cut-off values of the factors, like NLR?

Response: We did not choose the cut-off values, we only reported descriptive statistics on our subgroups.

Q7. The conclusion is superficial. Indeed, it is evident that the inflammatory status, such as NLR, MLR, PNI, AISI, and SII, hold significant relevance in the management of patients undergoing surgical treatment for colorectal cancer. However, how did we use the standard?

Response:Thank you very mush for the suggestion, we did make significant modification to our conclusion. There are no standard values yet since no consensus is reported in the scientific literature, we tested how it is a work on our sample. It is also valuable to secondary studies, such as meta-analysis

Q8. It’s suggested to check the grammar and typo errors.

Response: Thank you for your observation. We carefully proofread English with a support of a native English speaker.

Reviewer 3 Report

Comments and Suggestions for Authors

Authors performed a retrospective study spanning 7 years to study the effects of proinflammatory states on the progression of colorectal cancers by examining various inflammation markers including NLR, PLR, MLR, AISI, SIRI and SII. They successfully showed a statistically significant association between unfavorable prognosis and an increment in NLR, PLR, AISI or SII, while MLR or SIRI showed no statistically significant effect on prognosis. Although the approach is a common one, the data are solid and interesting especially in that the findings regarding MLR and SIRI are different to those regarding NLR, PLR, AISI or SII. Lack of statistical significance regarding the effects of MLR and SIRI may not come from the insufficiency in the patient number because MLR and SIRI showed opposite tendency to NLR, PLR, AISI or SII.

If authors provide a deep discussion regarding this point, the value of the manuscript will be highly upgraded.

Major concerns:

Please add a description discussing the difference between the effects of NLR, PLR, AISI and SII and those of MLR and SIRI referring multiple other papers to make clear the unique points regarding MLR and SIRI.     

Author Response

Q1. Authors performed a retrospective study spanning 7 years to study the effects of proinflammatory states on the progression of colorectal cancers by examining various inflammation markers including NLR, PLR, MLR, AISI, SIRI and SII. They successfully showed a statistically significant association between unfavorable prognosis and an increment in NLR, PLR, AISI or SII, while MLR or SIRI showed no statistically significant effect on prognosis. Although the approach is a common one, the data are solid and interesting especially in that the findings regarding MLR and SIRI are different to those regarding NLR, PLR, AISI or SII. Lack of statistical significance regarding the effects of MLR and SIRI may not come from the insufficiency in the patient number because MLR and SIRI showed opposite tendency to NLR, PLR, AISI or SII.

If authors provide a deep discussion regarding this point, the value of the manuscript will be highly upgraded.

Q2. Please add a description discussing the difference between the effects of NLR, PLR, AISI and SII and those of MLR and SIRI referring multiple other papers to make clear the unique points regarding MLR and SIRI.

Response: thank you very much we made serious modifications to the discussion part of the article. We hope that we have met your demands

Round 2

Reviewer 1 Report

Comments and Suggestions for Authors

thank you for the revised manuscript

Author Response

thank you!

Reviewer 2 Report

Comments and Suggestions for Authors

The authors have addressed my concerns.

Comments on the Quality of English Language

Minor editing of English language required.

Author Response

thnak you very much!

Reviewer 3 Report

Comments and Suggestions for Authors

In the revised manuscript, authors have sufficiently addressed the concerns raised by the reviewer. Therefore, the current manuscript has been sufficiently improved to warrant publication in Diseases.

Author Response

Thank you very much!